# Effects of a Senior Musical Program on the Physical Function and Cognitive Abilities of Older Women in the Community

**DOI:** 10.3390/healthcare11081174

**Published:** 2023-04-19

**Authors:** Byeong-Soo Kim, Ji-Youn Kim, Sam-Ho Park, Myung-Mo Lee

**Affiliations:** 1Department of Physical Therapy, Graduate School, Daejeon University, 62, Daehak-ro, Dong-gu, Daejeon 34520, Republic of Korea; 2Department of Performing Arts Contents, Daejeon University, 62, Daehak-ro, Dong-gu, Daejeon 34520, Republic of Korea; 3Department of Physical Therapy, Daejeon University, 62, Daehak-ro, Dong-gu, Daejeon 34520, Republic of Korea

**Keywords:** cognition, female, music therapy, postural balance, respiratory function

## Abstract

Background: We aimed to investigate the effects of a community-based senior musical program on the cognitive and physical functions in older women. Method: Older women aged ≥65 years participating in a program at a community welfare center were randomized to experimental (n = 17) or control (n = 17) groups. The control group participated in singing and yoga classes offered at the welfare center, while the experimental group participated in a senior musical program consisting of vocal training, dancing, and breathing training. The effects of the 12-week program (120 min/session, two sessions/week) and the intergroup differences in outcomes were compared using the cognitive impairment screening test (CIST), pulmonary function test (PFT), respiratory muscle pressure test (RPT), and static and dynamic balance tests. Result: The experimental group showed significant post-intervention changes in CIST scores, cardiorespiratory parameters, and static and dynamic balance (*p* < 0.05), but the control group only showed significant changes in some respiratory and balance parameters (*p* < 0.05). In comparison with the control group, the experimental group showed significantly greater post-intervention changes in the CIST score, PFT and RPT parameters, static balance, and Y-balance anterior (*p* < 0.05). Conclusions: The senior musical program enhanced older women’s cognitive, respiratory, and physical functions and promoted a sense of accomplishment and self-satisfaction.

## 1. Introduction

In 2021, the Republic of Korea was designated as an aged society with the older adult population reaching 16.5% of the total population owing to improved standards of living and advances in healthcare technology. At this pace, the country is projected to become a super-aged society with 20.3% of its population being older adults by 2025 [1]. With the expanding older adult population, the prevalence of geriatric diseases has been increasing continually, and older adults’ health problems have emerged as a critical societal issue in an aging society. In particular, the higher percentage of older women in comparison with older men has resulted in a high prevalence of chronic diseases and increased costs for healthcare and welfare [1], including the burden of social welfare costs, which may potentially be transferred to future generations [2].

Aging entails mental and physical functional decline [3] as well as problems in cognitive [2], cardiorespiratory [4], and musculoskeletal [5] functions. The classic geriatric diseases include dementia [6], chronic obstructive pulmonary disorder (COPD) [7], and pneumonia [8] and impaired balance may increase the risk for falls [9] and thus cause musculoskeletal injuries. These geriatric illnesses pose a significant threat to the health of older adults and are likely to result in death [10]. Furthermore, the onset of these diseases can lead to a rapid decline in both physical and economic quality of life, having a profound impact not only on the individuals but also their families [11].

Many community-based programs have been implemented to prevent geriatric diseases and improve the quality of life in older adults, and the effectiveness of these programs have been documented in the literature. Studies have proposed memory training [12], exercise therapy [13], and combined cognitive training [14] to prevent cognitive decline, and other studies have shown the effects of deep breathing exercise training [15], respiratory resistance training [16], and vibration stimulation training [17] in preventing cardiopulmonary decline. In addition, balance exercise [18], strength training [19], and aerobic exercise [20] have been introduced to prevent musculoskeletal injuries. Although these studies have demonstrated efficacy in addressing the underlying factors of geriatric diseases, a common challenge is that failing to engage participants or tailoring the level of difficulty for the participants may cause the participants to have difficulty participating in the intervention and experience low satisfaction [21].

In particular, Korean older women have a higher rate of dementia than male senior citizens [22], and pulmonary function decline occurs more frequently due to thoracic kyphosis [23]. In addition, although the rate of depression and falls is significantly higher, older women have a longer lifespan, so more research on maintaining a healthy life for the older women is needed [24,25].

In response to the limitations in the existing literature, there has been a growing interest in developing and implementing community-based programs aimed at enhancing participant engagement and satisfaction. Senior-targeting programs such as dancing [26], playing instruments [21], and choir participation [27] have been reported to have a positive influence on physical functioning as well as interest and self-satisfaction. However, previous studies have primarily focused on functional training and have not fully explored the diverse effects of such interventions. Thus, there is a need for new programs that can build upon the findings of previous studies and produce more multifaceted benefits.

In this context, this study aimed to develop and evaluate the effects of a senior musical program on older women’s cognitive, respiratory, and musculoskeletal functions, which play a key role in the development of geriatric diseases, and examine the clinical utility of this program as a community program for older women.

## 2. Materials and Methods

### 2.1. Participants

Ninety-two older women who participated in programs offered at the senior welfare center in the city of D were recruited. The inclusion criteria were female sex and age ≥ 65 years; independent ambulation without the use of assistive devices; cognitive impairment screening test (CIST) score above the cutoff (Table 1) [25]; and no surgical history for musculoskeletal disorders and no history of treatment for a chronic respiratory condition in the past two months. Individuals who had been diagnosed with a cardiocerebrovascular disease in the past six months or by a specialist, individuals with a history of hospitalization due to a respiratory disorder, individuals who use an assistive device for moving, and individuals who had difficulty in communication due to hearing or vocal cord impairment were excluded.

The purpose, procedure, and method of the study were explained to all participants, and only those who voluntarily consented to participate were enrolled. This study was approved by the bioethics committee of Daejeon University and is registered in the WHO International Clinical Trials Registry Platform (KCT0007769).

### 2.2. Procedure

This study is a preliminary study. This study used a two-group pre-test and post-test study design. Sample size was determined using the G*power program (ver. 3.1.9.2; University of Kiel, Kiel, Germany), and with reference to the main effect size (d) of 1.11 reported by Kim et al. [23], significance level (α) of 0.05, and power (1-β) of 0.8, the sample size was calculated to be 14 for each group. To account for a 15% dropout rate, we aimed to recruit at least 17 participants for each group. Of the 92 candidates who volunteered for the study, 15 who were under 64 years of age, 20 older men, and 23 with a CIST score below the cutoff were excluded, resulting in a total of 34 participants. Using a randomization program, these 34 participants were randomized to the experimental group (n = 17) to undergo the senior musical program or the control group (n = 17) to undergo singing and yoga classes offered by the welfare center.

The control group participated in singing and yoga classes offered at the welfare center. The singing class was led by an instructor, and the participants sang along to songs with recorded music playing in the background. A song was chosen every week for repeat practice until the participants could sing along without looking at the lyrics. The yoga class was led by an instructor, and the difficulty level was progressively increased over the weeks. Both the singing and yoga classes were conducted for 12 weeks (one hour/session, twice/week).

The experimental group underwent a senior musical program designed as a purposeful and motivational program that comprehensively enhances cognitive, respiratory, and musculoskeletal functions (Table 2). With reference to the study by Godoy et al. [28] and Suzuki et al. [29], the senior musical program included vocal training-based breathing exercise, a dance component tailored to meet the level of difficulty and metabolism appropriate for older adults, and songs arranged to allow for easy memorization. A five-minute warmup and cooldown stretching exercise was added before and after the intervention, respectively. The senior musical program was provided for 12 weeks (120 min/session, twice/week). The musical program was led by an instructor. To engage the participants and promote a sense of accomplishment, the participants’ families and friends were invited to their musical performance on the last day of the intervention (Figure 1).

The effects of the community-based health-promoting programs for older adults were compared using the CIST, pulmonary function test (PFT), respiratory pressure test (RPT), and the static and dynamic balance tests. One participant in the experimental group and one participant in the control group withdrew from the study midway due to personal reasons, so data from a total of 16 participants in each group were analyzed. After the 12-week program, the control group was offered the same musical program provided to the experimental group. The study design is illustrated in Figure 2.

### 2.3. Instruments

#### 2.3.1. Cognitive Impairment Screening Test

The CIST was developed by the Korean Ministry of Health and Welfare to address the limitations associated with the repeated use of existing cognitive impairment screening tools in the long term. It can screen for cognitive impairment and measure cognitive function in a relatively brief period of time (approximately 10 min), and cognitive impairment can be determined according to the participant’s age and education level (Table 1). The CIST involves a question-and-answer format between the participant and the examiner and consists of 13 questions across six domains (perception, attention, visuospatial function, executive function, memory, and language function). The total score is calculated by summing the scores for each item, and the total score ranges from 0 to 30. A higher score indicates better cognitive function, but the interpretation should be based on standards specific to the subject’s age and education [30].

#### 2.3.2. Pulmonary Function Test

In this study, we used the PFT to assess pulmonary function (Figure 3). The PFT is used to evaluate the functioning of the lungs and airways by measuring airflow and velocity to identify any medical abnormalities in the lungs and airways and respiratory health. We used a PFT device (MicroQuark, Cosmed, Italy) to determine whether the intervention program influenced pulmonary function [31]. The outcome measures included forced voluntary capacity (FVC), forced expiratory volume in 1 s (FEV1%), and maximal voluntary ventilation (MVV).

#### 2.3.3. Respiratory Pressure Test

The effects of the intervention program on respiratory muscle strength were measured using RPT [32]. The pressure of the airflow generated during inhalation and exhalation was indirectly measured using a respirometer (Micro RPM; Carefusion Ltd., Wokingham, UK) and expressed as respiratory muscle strength (cmH_2_O). The peak values of the pressure recorded during three forced inhalations and exhalations were recorded as the maximal inspiratory pressure (MIP) and maximal expiratory pressure (MEP), respectively.

#### 2.3.4. Static Balance Test

The effects of the intervention program on the participants’ static balance were quantitatively analyzed using a Wii balance board (WBB; Nintendo, Kyoto, Japan). The WBB is an affordable and portable force plate that can be used with the Balancia software to measure body sway by quantifying path length, velocity, and 95% area of the center of pressure (CoP). It shows high concurrent validity in comparison with a force plate (intraclass correlation coefficient (ICC) = 0.94) and is widely used in various clinical studies due to its portability and ease of use. With reference to the method used in a previous study on older adults, the participants were instructed to maintain a static standing position for 20 s with their eyes closed and then open, with their feet shoulder-width apart. The average value of three measurements was recorded [33].

#### 2.3.5. Dynamic Balance Test

Dynamic balance was tested using the Y-balance test. The Y-balance test is a tool used to assess dynamic balance by measuring the maximum distance a person can extend one leg to the anterior, posterior lateral (PL), and posterior medial (PM) directions from a marked Y line while maintaining balance with their dominant leg. It is a widely used clinical tool with a high reliability (ICC range: 0.85 to 0.93) [34].

### 2.4. Analysis

The collected data were entered into a Microsoft Excel sheet and analyzed using the IBM SPSS Win ver. 25.0. The data were presented using descriptive statistics, namely mean and standard deviation, and normality of data was tested using the Shapiro–Wilk test. Homogeneity of participants’ general characteristics and baseline outcome measures was analyzed with an independent t-test, and changes in the outcome measures after the intervention in comparison with the baseline within each group were analyzed with paired t-tests. The statistical differences in the intervention period between the two groups were analyzed with an independent-samples t-test. The statistical significance (α) was set at *p* < 0.05.

## 3. Results

Table 3 shows the participants’ general characteristics, namely age, sex, body weight, and height. The two groups were homogeneous in terms of their general characteristics at the baseline.

### 3.1. Cognitive Impairment Screening Test

The two groups did not show significant differences in their CIST score at the baseline, and only the experimental group showed a significant change in the CIST score after the intervention (*p* < 0.05; Table 4).

### 3.2. Pulmonary Function and Respiratory Pressure Tests

The two groups did not significantly differ in their PFT scores at the baseline, and the experimental group showed significant changes in FVC, FEV1, MVV, and MIP after the intervention in comparison with the baseline scores (*p* < 0.05). In contrast, the control group only showed a significant change in MVV (*p* < 0.05). Furthermore, the experimental and control groups showed significant differences in FVC, FEV1, and MIP (*p* < 0.05; Table 5).

### 3.3. Static Balance Test

The two groups did not show significant differences in the baseline static balance test score, while the experimental group showed significant changes in all parameters after the intervention (*p* < 0.05). In addition, the control group showed significant changes in velocity with eyes open (EO), path length with eyes closed (EC), and Area 95% EO after the intervention (*p* < 0.05). The experimental group showed significantly greater post-intervention changes than the control group in all parameters of the static balance test (*p* < 0.05; Table 6).

### 3.4. Dynamic Balance Test

The two groups did not show significant differences in the dynamic balance test score at the baseline, and the experimental group showed significant changes in all parameters after the intervention (*p* < 0.05). The control group showed significant changes only in the Y-balance test anterior value (*p* < 0.05). The two groups showed significant differences in the Y-balance test anterior and Y-balance test posterior lateral values (*p* < 0.05; Table 7).

## 4. Discussion

This study aimed to develop a senior musical program that can comprehensively promote respiratory and musculoskeletal functions in older adults and investigate its effects on older women’s cognitive, respiratory, and musculoskeletal functions. The results showed that the senior musical program positively improved older women’s cognitive and respiratory functions and their static and dynamic balance (*p* < 0.05).

Cognitive decline leads to dementia in older adults, thereby tremendously impairing their own and their family’s quality of life [35]. In a study on older adults with mild cognitive impairment, Suzuki et al. [26] instructed older adults to perform a dual task, which required them to do a calculation and move their body at the same time, and reported that these older adults showed significant changes in their cognitive abilities. Our findings supported these results, since both groups showed significant changes in their CIST score after their respective interventions (*p* < 0.05). We speculate that a dual task involving saying verbal lines and dancing simultaneously led to a greater activation of the cerebral cortex, thereby helping to improve cognitive functions.

To analyze the effects of the senior musical program on respiratory function, we examined the changes in the PFT and RPT scores. Eley and Gorman [32] administered a 26-week (60 min/week) wind instrument program to native Australians and found a 4% improvement in FVC, 6% improvement in FEV1, and 10% improvement in MVV, all of which were statistically significant. Furthermore, Fu M.C. et al. [33] administered a 12-week (75 min/week) breathing training and choir program and found a 20% improvement in MIP. In our PFT results, the experimental group that participated in the senior musical program showed a 13.9% improvement in FVC, 12.6% improvement in FEV1, and 14.4% improvement in MVV (*p* < 0.05). As for the RPT, the experimental group showed a 29.9% improvement in MIP (effect size 0.94) (*p* < 0.05). The potential motivation and sense of fulfillment experienced by the participants in the senior musical program may have contributed to their voluntary practice of the program’s exercises outside of the prescribed intervention, subsequently resulting in improved respiratory function. Nevertheless, no significant changes were observed in MEP after the intervention (effect size = 0.58), suggesting that the senior musical program had a limited impact on expiratory muscle strength because the activity involves singing while breathing in.

Falls caused by weakened lower-extremity muscles have been identified as a major cause of death among older adults. Thus, balance, which is based on muscle strength, is an important component in the quality of life of older adults [36]. Wallmann et al. [34] implemented a senior jazz dance program for middle-aged and older women aged 65 years and over and reported that their static balance was improved by 15%. Furthermore, Pattanasin Areeudomwong et al. [37] implemented a Muay Thai dance program for older adults aged 65 years and over and reported a significant improvement in their static and dynamic balance. Similarly, the experimental group in our study showed significant improvements in velocity EC (33.7%), EO (36.2%), path length EC (13.8%), EO (14.6%), area (95%), EC (22.8%), and EO (29.4%) after the intervention (*p* < 0.05). As for dynamic balance, the experimental group showed a 5.9% improvement in the Y-balance test anterior, 6.2% improvement in the Y-balance test PL, and 4.5% improvement in the Y-balance test PM values (*p* < 0.05). The improved balance ability observed in order women participants of the musical program may be due to the continuous dancing and singing required by the program, as well as the participants’ baseline respiratory function and low lower extremity muscle strength. The participants’ increased attention to balance during the program may have contributed to the observed outcomes.

This study had a few limitations that may impact the validity of the study. First, since the participants were allowed to participate in other programs offered at other welfare centers, it is possible that these programs also influenced the participants’ physical and cognitive functions. Second, the participants were recruited without regard to their income level and education level, so there were no different levels of difficulty available. In addition, the CIST evaluation tool is an evaluation tool developed according to the situation in Korea, and it must be changed to another cognitive function evaluation tool to apply it to other countries. Third, no follow-up assessments were performed, which precluded evaluations of the long-term effects. Fourth, as this study is a preliminary study, it is necessary to closely analyze the intervention effect through follow-up studies. Further research is needed to address these limitations and provide evidence to support the efficacy of senior musical programs.

## 5. Conclusions

This study aimed to develop and evaluate the effects of a senior musical program that may prevent physical and cognitive functional decline in older women. Our results showed that the senior musical program had positive effects on older women’s physical and cognitive functions. We recommend community welfare centers to offer this senior musical program to older women and encourage further follow-up studies to build on these findings.

## Figures and Tables

**Figure 1 healthcare-11-01174-f001:**
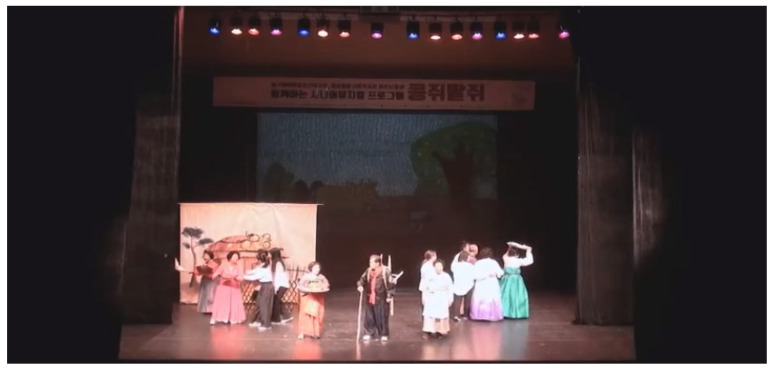
Senior musical program.

**Figure 2 healthcare-11-01174-f002:**
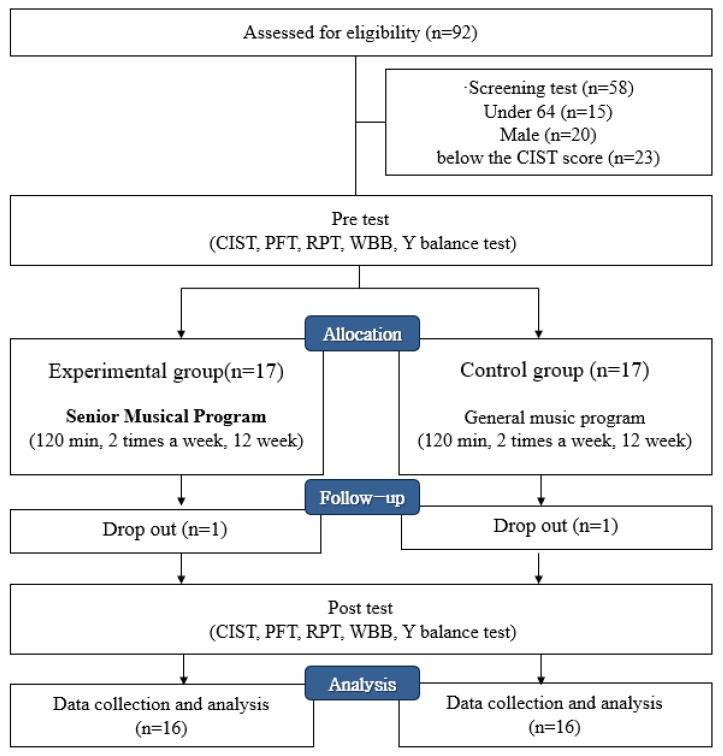
Flow chart of the study.

**Figure 3 healthcare-11-01174-f003:**
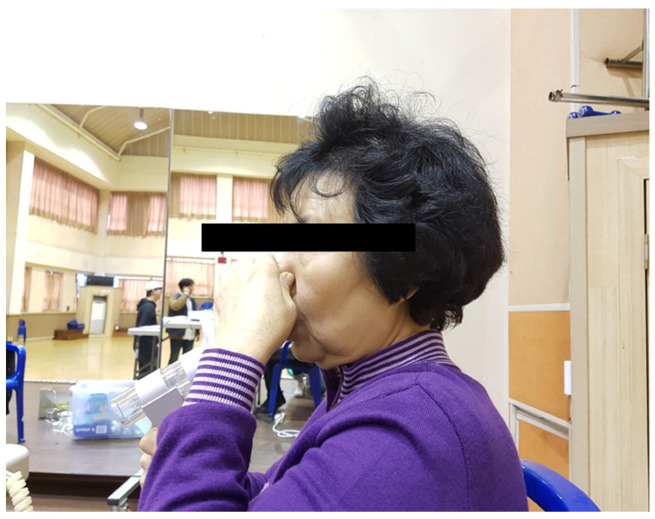
Pulmonary function testing.

**Table 1 healthcare-11-01174-t001:** Cognitive impairment screening test baseline scores.

Age (Years)	Education			
	Elementary School Graduation	Middle School Graduation	High School Graduation	University Graduation
50–59	22	24	26	27
60~69	21	23	25	26
70~79	19	22	22	25
80~89	16	18	20	22

**Table 2 healthcare-11-01174-t002:** Senior musical program.

Time	0~5 min	5~35 min	35~45 min	45~65 min	65~85 min	85~115 min	115~120 min
1	Warmup	Vocal training 1	Resting	Memorize the script 1	Learn choreography 1	Integrated practice 1	Cooldown
2	Warmup	Vocal training 2	Resting	Memorize the script 2	Learn choreography 2	Integrated practice 2	Cooldown
3	Warmup	Vocal training 3	Resting	Memorize the script 3	Learn choreography 3	Integrated practice 3	Cooldown
4	Warmup	Vocal training 4	Resting	Memorize the script 4	Learn choreography 4	Integrated practice 4	Cooldown
5	Warmup	Vocal training 5	Resting	Memorize the script 5	Learn choreography 5	Integrated practice 5	Cooldown
6	Warmup	Vocal training 6	Resting	Memorize the script 6	Learn choreography 6	Integrated practice 6	Cooldown
7	Warmup	Vocal training 7	Resting	Memorize the script 7	Learn choreography 7	Integrated practice 7	Cooldown
8	Warmup	Vocal training 8	Resting	Memorize the script 8	Learn choreography 8	Integrated practice 8	Cooldown
9	Warmup	Vocal training 9	Resting	Memorize the script 9	Learn choreography 9	Integrated practice 9	Cooldown
10	Warmup	Vocal training 10	Resting	Memorize the script 10	Learn choreography 10	Integrated practice 10	Cooldown
11	Warmup	Vocal training11	Resting	Memorize the script 11	Learn choreography 11	Integrated practice 11	Cooldown
12	Final performance						

**Table 3 healthcare-11-01174-t003:** General characteristics of the participants.

Variables	Experimental (n = 16)	Control (n = 16)	t	*p*
Age (years)	76.27 ± 2.72 *	73.90 ± 2.84	1.952	0.066
Height (cm)	158.91 ± 4.91	159.10 ± 4.93	−0.051	0.960
Weight (kg)	57.50 ± 4.44	57.60 ± 5.01	−0.048	0.962
BMI (kg/m^2^)	22.76 ± 1.77	22.72 ± 1.29	0.051	0.960
CIST (score)	22.18 ± 1.40	22.30 ± 1.57	−0.182	0.857

* Values are presented as mean ± SD. BMI: body mass index, CIST: cognitive impairment screening test.

**Table 4 healthcare-11-01174-t004:** Comparison of CIST score within and between groups.

	Variables	Experimental (n = 16)	Control (n = 16)	t (*p*)
CIST (score)	Pre	22.18 ± 1.40 *	22.30 ± 1.57	−0.182 (0.857)
	Post	25.27 ± 1.90	22.50 ± 1.27	
	Pre-post	−3.09 ± 1.7	−0.2 ± 0.92	4.773 (0.000) *^a^*
	t (*p*)	−6.029 (0.000) *^a^*	−0.688 (0.509)	
	Effect size (*d*)	1.85	0.14	

* Values are presented as mean ± SD. *^a^ p* < 0.05; CIST: cognitive impairment screening test; *d*: Cohen’s D.

**Table 5 healthcare-11-01174-t005:** Comparison of cardiopulmonary function within and between groups.

Variables		Experimental (n = 16)	Control (n = 16)	t (*p*)
FVC (L)	Pre	2.92 ± 0.56 *	2.16 ± 0.45	−3.403 (0.375)
	Post	3.39 ± 0.59	2.22 ± 0.40	
	Pre-post	−0.46 ± 0.31	−0.057 ± 0.15	3.79 (0.002) *^a^*
	t (*p*)	−5.00 (0.001) *^a^*	−1.21 (0.254)	
	Effect size (*d*)	0.82	0.14	
FEV1 (L)	Pre	2.64 ± 0.49	1.93 ± 0.46	3.347 (0.778)
	Post	2.98 ± 0.49	2.04 ± 0.46	
	Pre-post	−0.34 ± 0.24	−0.10 ± 0.26	2.13 (0.046) *^a^*
	t (*p*)	−4.73 (0.001) *^a^*	−1.34 (0.213)	
	Effect size (*d*)	0.69	0.23	
FEV1% (%)	Pre	90.74 ± 5.51	88.58 ± 6.07	0.858 (0.401)
	Post	88.69 ± 8.33	91.51 ± 10.48	
	Pre-post	2.05 ± 9.67	−2.93 ± 10.75	−1.12 (0.277)
	t (*p*)	0.70 (0.498)	−0.86 (0.411)	
	Effect size (*d*)	−0.29	0.34	
MVV (L)	Pre	61.70 ± 9.90	55.13 ± 13.86	1.259 (0.223)
	Post	72.04 ± 10.19	63.44 ± 15.58	
	Pre-post	−10.35 ± 6.12	−8.31 ± 6.31	0.075 (0.462)
	t (*p*)	−5.60 (0.000) *^a^*	−4.16 (0.002) *^a^*	
	Effect size (*d*)	1.03	0.56	
MIP (cmH_2_O)	Pre	−70.00 ± 31.01	−59.53 ± 23.19	−0.869 (0.396)
	Post	−99.82 ± 32.41	−56.66 ± 24.25	
	Pre-post	29.81 ± 33.20	-2.87 ± 8.74	−3.15 (0.009) *^a^*
	t (*p*)	2.98 (0.014) *^a^*	−1.04 (0.326)	
	Effect size (*d*)	−0.94	0.12	
MEP (cmH_2_O)	Pre	57.27 ± 25.99	57.42 ± 23.28	−0.014 (0.989)
	Post	73.07 ± 28.27	55.90 ± 24.72	
	Pre-post	−15.80 ± 39.64	1.52 ± 3.70	1.44 (0.179)
	t (*p*)	−1.32 (0.216)	1.30 (0.227)	
	Effect size (*d*)	0.58	−0.06	

* Values are presented as mean ± SD. *^a^ p* < 0.05, CIST: cognitive impairment screening test. FVC: forced vital capacity, FEV1: forced expiratory volume in one second, MVV: maximum voluntary ventilation, MIP: maximum inspiratory pressure, MEP: maximum expiratory pressure; *d*: Cohen’s D.

**Table 6 healthcare-11-01174-t006:** Comparison of Static balance test within and between groups.

Variables			Experimental (n = 16)	Control (n = 16)	t (*p*)
Velocity (cm/s)	EC	Pre	6.80 ± 1.42	6.63 ± 1.40	0.289 (0.776)
		Post	4.51 ± 1.31	6.72 ± 1.62	
		Pre-post	2.29 ± 1.44 *	−0.09 ± 0.48	−5.18 (0.000) *^a^*
		t (*p*)	5.28 (0.000) *^a^*	−0.62 (0.552)	
		Effect size (*d*)	−1.68	0.06	
	EO	Pre	5.49 ± 1.34	5.06 ± 1.13	0.810 (0.428)
		Post	3.50 ± 0.49	5.42 ± 1.13	
		Pre-post	1.20 ± 1.51	−0.37 ± 0.36	−4.99 (0.000) *^a^*
		t (*p*)	4.34 (0.001) *^a^*	−3.18 0 (0.011) *^a^*	
		Effect size (*d*)	−1.97	0.32	
Path length (cm)	EC	Pre	133.16 ± 18.35	136.39 ± 17.90	−0.408 (0.688)
		Post	114.80 ± 10.06	140.24 ± 19.08	
		Pre-post	18.36 ± 14.74	−3.84 ± 4.85	−4.72 (0.000) *^a^*
		t (*p*)	4.13 (0.002) *^a^*	−2.50 (0.033) *^a^*	
		Effect size (*d*)	−1.24	0.21	
	EO	Pre	117.34 ± 13.47	118.38 ± 15.59	0.810 (0.428)
		Post	100.17 ± 8.55	120.20 ± 17.76	
		Pre-post	17.17 ± 12.24	−1.82 ± 5.36	−4.68 (0.000) *^a^*
		t (*p*)	4.65 (0.001) *^a^*	−1.07 (0.312)	
		Effect size (*d*)	−1.52	0.11	
Area 95% (cm^2^)	EC	Pre	26.03 ± 6.43	26.14 ± 4.75	−0.043 (0.966)
		Post	20.10 ± 6.80	27.55 ± 3.91	
		Pre-post	5.92 ± 4.16	−1.42 ± 2.96	−4.61 (0.000) *^a^*
		t (*p*)	4.72 (0.001) *^a^*	−1.51 (0.165)	
		Effect size (*d*)	−0.90	0.32	
	EO	Pre	11.11 ± 1.77	12.45 ± 2.67	−1.368 (0.187)
		Post	7.84 ± 1.84	12.87 ± 2.78	
		Pre-post	3.27 ± 2.08	−0.42 ± 0.39	−5.77 (0.000) *^a^*
		t (*p*)	5.21 (0.000) *^a^*	3.392 (0.008) *^a^*	
		Effect size (*d*)	−1.81	0.15	

* Values are presented as mean ± SD. *^a^ p* < 0.05, EO: eyes open, EC: eyes closed; *d*: Cohen’s D.

**Table 7 healthcare-11-01174-t007:** Comparison of Y-balance test results within and between groups.

	Variables		Experimental (n = 16)	Control (n = 16)	t (*p*)
Y-balance test (cm)	Anterior	Pre	65.36 ± 8.68	67.80 ± 8.50	−0.649 (0.524)
		Post	69.49 ± 8.11	65.40 ± 7.85	
		Pre-post	−4.13 ± 2.73 *	2.4 ± 2.55	5.645 (0.000) *^a^*
		t (*p*)	−5.010 (0.001) *^a^*	2.979 (0.015) *^a^*	
		Effect size (*d*)	0.49	−0.29	
	Posterior Lateral	Pre	78.09 ± 10.15	66.30 ± 10.23	2.649 (0.166)
		Post	83.29 ± 10.42	66.00 ± 9.98	
		Pre-post	−5.2 ± 2.8	0.3 ± 4.85	3.219 (0.005) *^a^*
		t (*p*)	−6.157 (0.000) *^a^*	0.195 (0.849)	
		Effect size (*d*)	0.51	−0.03	
	Posterior Medial	Pre	71.73 ± 8.11	67.80 ± 13.63	0.812 (0.427)
		Post	75.08 ± 6.92	67.00 ± 9.98	
		Pre-post	−3.35 ± 3.59	0.8 ± 8.48	1.487 (0.153)
		t (*p*)	−3.095 (0.011) *^a^*	0.298 (0.772)	
		Effect size (*d*)	0.44	−0.067	

* Values are presented as mean ± SD. *^a^ p* < 0.05, CIST: cognitive impairment screening test; *d*: Cohen’s D.

## Data Availability

Not applicable.

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
