# Peer review of "Effects of a Senior Musical Program on the Physical Function and Cognitive Abilities of Older Women in the Community"

_healthcare, 2023, doi:10.3390/healthcare11081174_

Round 1

Reviewer 1 Report

- Please explain why you did not use p-value adjustments in multiple t-tests

- It would be helpful to also provide effect sizes for each result

Author Response

#1 Please explain why you did not use p-value adjustments in multiple t-tests

Response: Thank you for your valuable comments. This study is a control group pre-post study design, so multiple t tests are not required, and independent t tests and paired t tests are sufficient for statistical analysis.

#2 It would be helpful to also provide effect sizes for each result

Response: Thank you for your pointing. As you suggested, we presented the effect size for each results.

Reviewer 2 Report

Please avoid using the word "elderly" manuscript.

Abstract: The abstract does not follow the journal's structure. Please rewrite it to adapt to this format (background, methods, results and conclusion.

Introduction:

Please explain in more depth and/or provide a reference for what was explained in line 56-59

Material and Methods:

Please explain the exclusion criterion, specially male exclusion

Explain the few number of patients in the study

I don’t understand the interest of figure 1 and 3

The use of this CIST test makes the findings in this area not easily generalizable to the population of other countries. Justify its use (line 138-148) and comment on its limitation (285-292).

Results

Explain more the results, not only the p value

Disccusion

A 3 points difference in pre-post examination in CIST, is clinically relevant? In some findings, a difference of 3 points can be significant, such as in the Short Physical Performance Battery (SPPB), because the scale used for measurement is more sensitive and can detect small changes in physical function. However, in other findings, such as blood pressure, a difference of 3 points may not be significant because the scale used for measurement is less sensitive and small changes may not have a noticeable impact on health outcomes. Explain for the rest of the differences

Conclusion

With the small sample size, the doubts about significant changes, and absence of follow-up in the patients, the conclusions are very optimistic.

Author Response

#1 Please avoid using the word "elderly" manuscript.

Response: Thank you for your valuable comments. As you suggested, we modified it to older women.

#2 Abstract: The abstract does not follow the journal's structure. Please rewrite it to adapt to this format (background, methods, results and conclusion.

Response: Thank you for your valuable comments. As you suggested, we modified the Abstract session.

#3 Introduction:

Please explain in more depth and/or provide a reference for what was explained in line 56-59

Although these studies have demonstrated efficacy in addressing the underlying factors of geriatric diseases, a common challenge is that failing to engage participants or tailoring the level of difficulty for the participants may cause the participants to have difficulty participating in the intervention and experience low satisfaction.

Response: Thank you for your valuable comments. As you suggested, Added a reference that can explain the content.

#4 Material and Methods:

Please explain the exclusion criterion, specially male exclusion

Response: Thank you for your valuable comments. We have added content to the introduction.

[In particular, the higher percentage of older women in comparison with older men has resulted in a high prevalence of chronic diseases and increased costs for healthcare and welfare, including the burden of social welfare costs, which may potentially be transferred to future generations.]

#5 Explain the few number of patients in the study

Response: Thank you for your valuable comments. It is written in the study inclusion and exclusion criteria.

[The inclusion criteria were female sex and age ≥ 65 years; independent ambulation without the use of assistive devices; cognitive impairment screening test (CIST) score above the cutoff; and no surgical history for musculoskeletal disorders and no history of treatment for a chronic respir-atory condition in the past two months.

Individuals who had been diagnosed with a cardiocerebrovascular disease in the past six months or by a specialist, individuals with a history of hospitalization due to a respiratory disorder, individuals who use an assistive device for moving, and individuals who had difficulty in communication due to hearing or vocal cord impairment were excluded.]

#6 I don’t understand the interest of figure 1 and 3

Response: Thank you for your valuable comments. It is attached to the research description for a little more amplification.

[Figure 1 is the musical performance by research participants participating in the Senior Musical Program and Figure 3 is a pulmonary function test, which is a representative evaluation of this study.]

#7 The use of this CIST test makes the findings in this area not easily generalizable to the population of other countries. Justify its use (line 138-148) and comment on its limitation (285-292).

Response: Thank you for your valuable comments. In the case of CIST, it is an evaluation tool developed to suit the situation in Ministry of health and welfare in South Korea. To apply it overseas, it has to be changed and applied to the situation abroad. This was noted in the review.

[The CIST was developed by the Korean Ministry of Health and Welfare to address the limitations associated with the repeated use of existing cognitive impairment screening tools in the long term]

#8 Results

Explain more the results, not only the p value

Response: Thank you for your valuable comments. We presented the effect size for each results.

#9 Disccusion

A 3 points difference in pre-post examination in CIST, is clinically relevant? In some findings, a difference of 3 points can be significant, such as in the Short Physical Performance Battery (SPPB), because the scale used for measurement is more sensitive and can detect small changes in physical function. However, in other findings, such as blood pressure, a difference of 3 points may not be significant because the scale used for measurement is less sensitive and small changes may not have a noticeable impact on health outcomes. Explain for the rest of the differences

Response: Thank you for your valuable comments.

CIST is a complex cognitive assessment tool that conducts a cognitive test through various items. In general, cognitive tests show characteristics that can come out low, but the fact that they came out high in the post-test in this study can be said to be clinically effective in cognitive improvement. For this, please refer to the reference of the CIST evaluation tool.

#10 Conclusion

With the small sample size, the doubts about significant changes, and absence of follow-up in the patients, the conclusions are very optimistic.

Response: Thank you for your pointing. This study investigates the effects of senior musical programs on the physical function and cognitive ability of elderly women in the community. Based on the main effect size results of previous studies, the minimum number of people recruited between groups was 17 for a total of 34 people. However, follow-up in the patients is a limitation of this study.

Reviewer 3 Report

Very good study and manuscript.

A few minor comments .

1. Line 104 the word "trot" I believe should be "to" so the phrase reads: "...san along to songs.."

2. For the PFT, Respiratory pressure, static and dynamic balance tests please provide possible score ranges and interpretation, that is, higher score means better or worse, and what is considered "normal" for age groups.  This is important in that there is always difficulty demonstrating improvement in a group with high functional abilities of any kind.  

3. It would be nice to see a table of all the demographics for the two groups, such as education, marital status, living independently or not.  If you did not collect this data then say so as if the two groups were not the same on this it could effect motivation and thus outcomes.

4. Line 234 please spell out "PL" as this is your first use of this abbreviation.

5. Discussion: Be mindful of overemphasizing statistical improvement over clinical or functional improvement as perceived by the older person.  Often these clinical tests do not really capture what is functionally important to our elderly.  A very good example is medication that statistically improves cognition according to some test, but realistically neither family nor patient see any difference in daily functionality.  Your interventions by their very nature of being more "real world" and valued by the participants has a much better chance of real and valued improvement by the participants.  In the future what really is the outcome desired? Fewer doctor visits, less medication use, fewer hospitalizations, improved years of independent function?  Remember to look beyond the end of the current study to set yourself up for the next one.

5. Limitations: add that your sample is quite small and limited to women. Another is the involvement of the families for the experimental group.  Social support outside the experimental conditions can impact outcomes as well.

General comments: Since you also gave the control group the same intervention after they did the control version, it is too bad you did not also collect your outcomes as this might have yielded data to demonstrate the strength of your intervention.  It would also have been interesting to see if the intervention changed any particular aspect/category of the CIST but your sample is too small.  Consider for future studies?

Author Response

#1 Line 104 the word "trot" I believe should be "to" so the phrase reads: "...san along to songs.."

Response: Thank you for your pointing. As you suggested, we have revised the text.

#2 For the PFT, Respiratory pressure, static and dynamic balance tests please provide possible score ranges and interpretation, that is, higher score means better or worse, and what is considered "normal" for age groups. This is important in that there is always difficulty demonstrating improvement in a group with high functional abilities of any kind. 

Response: Thank you for your valuable comments. Unfortunately, in the case of human pulmonary function and respiratory muscle strength, because the characteristics are configured differently for each individual, it has characteristics that cannot be generalized in a group study to say a normal category. The results of this study were statistically analyzed for the improvement of respiratory function, so please refer to it.

#3 It would be nice to see a table of all the demographics for the two groups, such as education, marital status, living independently or not.  If you did not collect this data then say so as if the two groups were not the same on this it could effect motivation and thus outcomes.

Response: Thank you for your valuable comments. The study was approved by the IRB on the condition that no sensitive personal information was collected at the time of approval. I did not collect the contents you said, so please forgive me for this.

#4 Line 234 please spell out "PL" as this is your first use of this abbreviation.

Response: Thank you for your pointing. As you suggested, we have revised the text.

#5 Discussion: Be mindful of overemphasizing statistical improvement over clinical or functional improvement as perceived by the older person.  Often these clinical tests do not really capture what is functionally important to our elderly.  A very good example is medication that statistically improves cognition according to some test, but realistically neither family nor patient see any difference in daily functionality.  Your interventions by their very nature of being more "real world" and valued by the participants has a much better chance of real and valued improvement by the participants.  In the future what really is the outcome desired? Fewer doctor visits, less medication use, fewer hospitalizations, improved years of independent function?  Remember to look beyond the end of the current study to set yourself up for the next one.

Response: Thank you I was impressed by your comment.

#6 Limitations: add that your sample is quite small and limited to women. Another is the involvement of the families for the experimental group.  Social support outside the experimental conditions can impact outcomes as well.

Response: This study investigates the effects of senior musical programs on the physical function and cognitive ability of elderly women in the community. Based on the main effect size results of previous studies, the minimum number of people recruited between groups was 17 for a total of 34 people.

#7 General comments: Since you also gave the control group the same intervention after they did the control version, it is too bad you did not also collect your outcomes as this might have yielded data to demonstrate the strength of your intervention.  It would also have been interesting to see if the intervention changed any particular aspect/category of the CIST but your sample is too small.  Consider for future studies?

Response: Thank you for your pointing. As you suggested, we will consider it for future research.

Round 2

Reviewer 2 Report

The article is eligible for publication after the modifications and comments made